# Understanding the Role of GLUT2 in Dysglycemia Associated with Fanconi–Bickel Syndrome

**DOI:** 10.3390/biomedicines10092114

**Published:** 2022-08-29

**Authors:** Sanaa Sharari, Basirudeen Kabeer, Idris Mohammed, Basma Haris, Igor Pavlovski, Iman Hawari, Ajaz Ahmad Bhat, Mohammed Toufiq, Sara Tomei, Rebecca Mathew, Najeeb Syed, Sabah Nisar, Selma Maacha, Jean-Charles Grivel, Damien Chaussabel, Johan Ericsson, Khalid Hussain

**Affiliations:** 1Division of Biological and Biomedical Sciences, College of Health and Life Sciences, Hamad Bin Khalifa University (HBKU), Qatar Foundation, Doha 34110, Qatar; 2Department of Pediatric Medicine, Division of Endocrinology, Sidra Medicine, Doha 26999, Qatar; 3Research Branch, Sidra Medicine, Doha 26999, Qatar; 4School of Medicine and Medical Science, University College Dublin, Belfield, 4 D4 Dublin, Ireland

**Keywords:** Fanconi–Bickel syndrome (FBS), dysglycemia, glucose transporter 2 (GLUT2), PBMCs (peripheral blood mononuclear cells), miRNAs

## Abstract

Fanconi–Bickel Syndrome (FBS) is a rare disorder of carbohydrate metabolism that is characterized by the accumulation of glycogen mainly in the liver. It is inherited in an autosomal recessive manner due to mutations in the *SLC2A2* gene. *SLC2A2* encodes for the glucose transporter GLUT2 and is expressed in tissues that are involved in glucose homeostasis. The molecular mechanisms of dysglycemia in FBS are still not clearly understood. In this study, we report two cases of FBS with classical phenotypes of FBS associated with dysglycemia. Genomic DNA was extracted and analyzed by whole-genome and Sanger sequencing, and patient PBMCs were used for molecular analysis. One patient had an exonic *SLC2A2* mutation (c.1093C>T in exon 9, R365X), while the other patient had a novel intronic *SLC2A2* mutation (c.613-7T>G). Surprisingly, the exonic mutation resulted in the overexpression of dysfunctional GLUT2, resulting in the dysregulated expression of other glucose transporters. The intronic mutation did not affect the coding sequence of GLUT2, its expression, or glucose transport activity. However, it was associated with the expression of miRNAs correlated with type 1 diabetes mellitus, with a particular significant overexpression of hsa-miR-29a-3p implicated in insulin production and secretion. Our findings suggest that *SLC2A2* mutations cause dysglycemia in FBS either by a direct effect on GLUT2 expression and/or activity or, indirectly, by the dysregulated expression of miRNAs implicated in glucose homeostasis.

## 1. Introduction

Fanconi and Bickel initially reported the clinical features of the eponymous syndrome (FBS) in 1949 [1]. Mutations in glucose transporter 2 (GLUT2) were first reported in three FBS patients, including the first patient in 1997 [2]. More than 100 FBS cases with different *SLC2A2* mutations (missense, nonsense, Fs/InDel, intronic, and compound heterozygous) have been reported to date [3,4,5,6,7,8,9]. The *SLC2A2* gene encodes for low-affinity-facilitated glucose transporter 2 (GLUT2, SLC2A2-201 ENST00000314251.8) [3,10]. GLUT2 is expressed in tissues that play a vital role in glucose homeostasis; GLUT2 in the intestine absorbs glucose from the diet and transports it to the blood [11,12].

GLUT2 in the human liver is considered to be a bidirectional transporter. It is involved in taking up glucose for storage as glycogen during the feeding state. It also plays a role in releasing glucose generated either by gluconeogenesis or glycogenolysis during fasting [13,14,15]. Fasting hypoglycemia, postprandial hyperglycemia, and glycogen storage in FBS patients can be explained by a disturbance in glucose transport and metabolism in the liver. In addition, in the kidney, GLUT2 releases the reabsorbed glucose back to the circulation. Previous investigations suggested that GLUT2 dysfunction in the kidney is associated with glycosuria and glycogen storage [16,17].

Moreover, GLUT2 is highly expressed in rat beta cells but not in humans. Animal studies suggested that GLUT2 in beta cells has a role in glucose uptake and insulin secretion. However, the role of GLUT2 in human beta cells is still unclear due to a lack of studies [6,18,19,20]. FBS patients develop different patterns of dysglycemia with different severity (fasting hypoglycemia, postprandial hyperglycemia, frank diabetes mellitus, transient neonatal diabetes, and gestational diabetes) regardless of the mutation type.

The molecular mechanisms of dysglycemia in FBS are still controversial and not well explained [3]. One recent study selected a group of FBS mutations associated with dysglycemia and expressed the corresponding GLUT2 mutant proteins in HEK293 cells. Cells transfected with mutated constructs showed the same fructose uptake activity as cells transfected with the wild-type GLUT2, except for p.Thr198Lys, which showed a slight decrease in fructose uptake activity [21].

In the current report, we aim to report two new cases of Fanconi–Bickel syndrome associated with dysglycemia. We also aim to better understand the mechanisms of dysglycemia in these two patients by performing functional studies in patient peripheral blood mononuclear cells (PBMCs).

## 2. Materials and Methods

### 2.1. Patient Recruitment and Clinical Information

This study included two patients diagnosed with FBS associated with dysglycemia. Patients and their families were recruited at Sidra Medicine, and blood samples were collected to perform genetic analysis and to extract PBMCs. In addition, two healthy controls (age- and gender-matched) were recruited at Sidra Medicine, and blood was withdrawn to extract PBMCs. The biochemical and radiological test results of the patients were collected from the medical information system at Sidra Medicine. All methods were performed in accordance with the relevant guidelines and regulations by the Declaration of Helsinki and approved by the Sidra Institutional Review Board (IRB) Committee for the protection of human subjects, Sidra Medicine. Written informed consent forms were completed by all family members involved in the study.

### 2.2. DNA, miRNA, RNA, and PBMCs Isolation

Genomic DNA was isolated from peripheral blood samples following the manufacture’s protocol (QIAamp DNA Blood Maxi Kit, Qiagen). PBMCs were purified using the following optimized protocols: (1) blood was diluted with an equal volume of serum-free RPMI media (with glutamine, penicillin/streptomycin); (2) 2 volumes of diluted blood were subsequently added to the side of a Falcon tube containing 1 volume Ficoll-Paque^TM^ PLUS (GE Healthcare); (3) the resulting solution was centrifuged at 2000 rpm for 30 min (acceleration 6, deceleration 0), (4) the PBMCs layer was recovered and washed twice with RPMI medium at 1500 rpm for 10 min (acceleration and deceleration 9); and (5) the supernatant was discarded and the cell pellet was resuspended in 1 mL RPMI for cells counting. PBMCs were either used immediately for glucose uptake assays or stored at −80 °C with QIAzol lysis reagent for a later extraction of miRNA and RNA according to the manufacture’s protocols (miRNeasy Mini kit, QIAGEN).

### 2.3. Whole-Genome and Sanger Sequencing

We performed whole-genome sequencing using the Illumina HiSeq platform at 30X coverage. The paired FASTQ files resulting from sequencing contained the nucleotides sequence reads and the quality scores for each read. Quality control on fastq files was performed using fastqc software. Sequence reads were aligned to the hs37d53 reference genome using bwa.kit (v0.7.12), which generated a bam file. Quality control on mapped reads was performed using Picard. The variant calling was performed following GATK (4.1) best practices, and joint calling was performed on samples together. The VCF file was normalized and left-aligned using vt. The functional annotation of the combined VCF files was performed using snpEff. The annotated VCF file was annotated further with Clinvar, Gnomad, Thousand Genome projects Allele frequencies, CADD scores, etc., using vcfanno. Each annotated file was uploaded to the Gemini database using vcf2db utilities, and Gemini was used to perform filtering population allele frequency based variant effect and family-based segregation analysis. DELLY software was used for copy number variations (CNVs). Sanger sequencing was used to confirm the mutation in patients and their parents using specific primers (Appendix A). Snapgene software was used for Sanger sequencing analysis.

### 2.4. Flow Cytometry

PBMCs were isolated from peripheral blood by centrifugation, as explained above. The optimized protocol was performed as follows: (1) preparation of stained samples: 1 million cells were stained in 99 µL Dulbecco’s Phosphate Buffered Saline (DPBS) with 1 µL Zombie UV Fixable Viability Dye (for live/dead discrimination) for 15 min in the dark, followed by washing with DPBS at 500× *g* for 5 min. Cells were incubated with either 10 µL of anti-human GLUT2 PE-conjugated mouse IgG2a antibody (R&D SYSTEMS, FAB14148) or with FMO only. An antibody cocktail of 5 µL CD4 (BV 421), 5 µL CD8 (BV 605), 5 µL CD14 (AF 647), and 7 µL CD19 (BV 650) was added to all tubes. Subsequently, brilliant staining buffer (BD Biosciences) was added to a final staining volume of 100 µL and incubated for 15 min in the dark. Washing was performed with cell-staining buffer by centrifugation at 500× *g* for 5 min. The cells were fixed with 4% PFA in PBS, incubated for 20 min in the dark at room temperature, and washed once with DPBS, by centrifugation for 5 min at 500× *g*. (2) Single-stain compensation was performed using UltraComp beads (ThermoFisher, Waltham, Massachusetts, United States) individually incubated with 5 µL of each antibody in the dark for 15 min, washed with DPBS, fixed with 4% PFA, and resuspended in DPBS. (3) A small portion of cells was kept unstained. The data were acquired on a BD FACS Symphony A5 flow cytometer. Data analysis was performed using FlowJo Software.

### 2.5. Western Blot

Protein lysates from the patient and control PBMCs were extracted using RIPA buffer with protease and phosphatase inhibitors to examine the expression of GLUT2 protein. Protein lysates were incubated with Laemmli buffer (4×) at 37 °C for 30 min. Subsequently, samples were run on 10-well 4–12% gels (ThermoFisher, Waltham, MA, USA) using MES buffer at 100 V for 90 min and transferred to a nitrocellulose membrane using wet transfer. The membranes were blocked in 5% milk for 2 h and incubated with primary GLUT2 antibody (20436-1-AP (Proteintech), 1:300) overnight, followed by five 5 min washes with TBST. The secondary anti-rabbit antibody was added for 2 h, followed by five 5 min washes with TBST, and developed using SuperSignal^TM^ West Pico PLUS Chemiluminescent Substrate. Images were collected using a ChemiDoc^TM^ MP Imaging system (BIO-RAD, Hercules, CA, USA). Beta-actin was used as an endogenous control. ImageJ was used to quantify the intensity of individual protein band.

### 2.6. Glucose Uptake Assay

Patient and control PBMCs were immediately used after extraction from fresh peripheral blood samples in glucose uptake assays. A total of 500,000 cells were starved in glucose-free RPMI medium for 1 h, followed by the addition of 10 µL of 10 mM 2-DG for 1 h. Cells were lysed with extraction buffer at 85 °C for 40 min. The reaction mixtures were neutralized with a 10 µL neutralizing buffer. The glucose uptake activity was tested as recommended by the manufacturer (Abcam, Cambridge, East Anglia, UK, ab136955), and the absorbance was measured at 412 nm using a microplate reader (Flaoster, Omega).

### 2.7. qRT-PCR

The expression of other glucose transporters (GLUT1 and GLUT3) in patients and control PBMCs was quantitatively assessed. The RNA was extracted from PBMCs as described above and normalized to 200 ng for cDNA synthesis. A total of 3 µL of cDNA products was added and amplified using a 20 µL reaction of SYBR^R^ green master mix at primer-specific Tms (Appendix A). The mRNA levels were measured on a QuantStudioTM 12K Flex SystemBlock 96-well instrument. Each PCR reaction used distilled water instead of cDNA as a negative control. Fluorescent data were acquired during the extension phase. Melting curves were generated at the end of each PCR reaction to verify primer specificities. Samples were run in technical replicates. Gene expression was calculated using the 2^-deltaCt method with GAPDH used as the reference gene. 

### 2.8. RNASeq Analysis

The RNA extracted from the samples of the patients and controls were submitted to the Genomics Core at Sidra Medicine for RNA-Seq analysis. Library preparation was performed using the TruSeq Illumina RNA Library Prep kit. Libraries were quantified using the KAPA HiFi Library quantification kit on a Roche LightCycler 480 (Roche, Basel, Switzerland). Cluster generation was performed on a cBot instrument. Samples were sequenced on Illumina HiSeq 4000. The processing of RNASeq data was performed using the bcbio rnaseq pipeline (bcbio version 1.2.3). Prior to alignment, a quality check of raw data was conducted using FastQC version 0.11.9. Alignment was performed using STAR (version 2.6.1d) and reads were mapped to the hg38 genome. Following alignment, Samtools 1.3 was used to collect metrics on BAM files, which were further used to generate a multiQC report. FeatureCounts (version 2.0.0) was used to estimate the expression counts of each gene. Sample read counts were adjusted for library size and normalized using the Trimmed Mean of M-values (TMM) method using Bioconductor package EdgeR (version 3.34.1). Data were log2 transformed. Fold Change (FC) was calculated between patients and their respective controls using EdgeR. Genes with log2 FC of +2 and −2 were used for downstream pathway analysis with Ingenuity Pathway Analysis (version 01-18-05) and the Bioconductor package Complex Heatmap (version 2.8.0). ggplot2 library (version 3.3.5) was used for visualization and plotting.

### 2.9. miRNA Analysis

The total RNA extracted from patient 1 (db-bl-0008), her mother, and age- and gender-matched control 1 were submitted to the Omics Core at Sidra Medicine for Nanostring miRNA profiling. The Nanostring miRNA panel v3b (including ~800 targets) was run on all samples. A total of ~150 ng of total RNA was used as input for each of the samples assessed. Sample preparation, ligation, hybridization, detection, and scanning were performed as per the manufacturer’s instructions. Following hybridization, the samples were transferred to the nCounter Prep Station, where excess probes were removed, and samples were aligned and immobilized on the nCounter cartridge. The cartridge was placed on the nCounter Digital Analyzer for data collection. The nSolver data analysis software (version 4.0 NanoString Technologies) was used for the assessment of QC and the normalization of the raw gene expression counts. We used the recommended default parameters for quality control flagging; briefly, flags were generated if samples did not meet the following QC criteria: imaging threshold with FOV registration of at least 75%, binding density between 0.05 and 2.25, positive control, ligation control linearity with R2 > 0.95, positive control limit of detection at 0.5 fM, and positive control > or = 2 standard deviations above the mean of the negative controls. Data are presented as normalized raw counts. Data were imported on ROSALIND (https://app.rosalind.bio/ (accessed on 10 June 2021) and Partek Genomic Suite (Partek, St. Louis, MI, USA) for secondary downstream analysis. Functional gene network analysis was performed using the Ingenuity Pathway Analysis system (QIAGEN, Hilden, Germany), which transforms large data sets into a group of relevant networks containing direct and indirect relationships between genes based on known interactions in the literature.

### 2.10. Statistical Analysis

*p*-values between groups were calculated using two-tailed t-tests. Error bars correspond to the average of three independent experiments except for qRTPCR, where they correspond to the average of technical replicates.

## 3. Results

### 3.1. Clinical and Genetic Information of Patients

Case 1 was a 19-year-old Pakistani female born to consanguineous parents with the classical phenotype of FBS (Figure 1a). She was born at term by normal vaginal delivery with a weight of 2.2 kg (<3rd centile). The patient developed her first symptom at 1.5 years old, and was diagnosed with FBS at 12 years old. At the age of 2 years, she developed rickets and recurrent fractures (Figure 1c). Upon examination, her height was 100 cm (<5th percentile) and her weight was 23 kg (<5th percentile) (Figure 1b).

In addition, the patient developed renal tubular acidosis with aminoaciduria and her biochemical findings showed dysglycemia (fasting hypoglycemia and postprandial hyperglycemia, low levels of C peptide) (Table 1). Her HbA1c level was high. Thus, she was diagnosed with diabetes mellitus at the age of 17 years. The diabetes mellitus type 1 evaluation was negative for all autoantibodies, and the patient did not have any features of insulin resistance. The HbA1c level decreased from 6.4% to 6.2% upon treatment with insulin and a DPP4 inhibitor. Moreover, the patient had hepatomegaly (Figure 1c) with impaired liver function tests (Table 1). The patient received multiple medications for electrolyte imbalance (sodium bicarbonate, potassium, phosphorous, and vitamin D).

The patient’s WGS revealed a novel homozygous mutation c.613-7T>G: IVS5-7T>G in intron 5 of the *SLC2A2* gene (NM_000340.1), and both parents were carriers. The mutation was confirmed by Sanger sequencing of DNA (Figure 2a). This mutation has not been reported before and is expected to change the splice site of exon 6, causing the addition of two amino acids by reducing the quality of the splice acceptor site in intron 5 and creating a new cryptic splice acceptor site upstream of the natural splice site. However, the patient’s cDNA Sanger sequencing revealed no extra nucleotides between exons 5 and 6 (Figure 2b), which means the mutation did not affect the coding region of GLUT2 protein. Appendix A shows the expected unchanged GLUT2 topology for the patient. 

Therefore, we aimed to investigate if there was another possible pathogenic mutation in the *SLC2A2* gene or other genes implicated that might impact GLUT2 expression. The whole-genome sequencing data were uploaded in SeqR, and one additional intronic mutation was observed in the *SLC2A2* gene c.15+816A>C (predicted to be likely benign) (Appendix A). The VCF file was generated, and only two non-pathogenic mutations (PIK3CD c.4C>A, a benign variant, and EP300 c.1169-4T>C, an intronic mutation) out of 60 genes that regulate GLUT2 expression were detected (Appendix A). Furthermore, we filtered all potential damaging exonic and splice mutations in SeqR, and we obtained 14 (*ACTL9- ATP8B4- BMPR1B- COL17A1- CRLF1- FAHD2A- GLDC- HAL- MBTPS2 -NTHL1- PRR5L- SIAE- SLIT3- TMOD3*) out of 757 genes; it is noteworthy that none of them were associated with FBS phenotypes.

Case 2 was an eight-year-old Sudanese boy born to consanguineous parents with the classical phenotype of FBS (Figure 3a). He was born full term by normal vaginal delivery with a weight of 3.5 kg. The patient developed his first symptom at 1 year old, and was diagnosed with FBS aged 1 year four months. The patient developed dysglycemia (fasting hypoglycemia, postprandial hyperglycemia, and diabetes mellitus) at two years of age. In addition to hypophosphatemia, hypokalemia, and hyponatremia, the biochemical investigations of the patient showed an increase in random glucose levels and a decrease in the levels of C peptide and insulin (Table 2). The patient’s dysglycemia was controlled by diet and the addition of cornstarch. This patient also had hepatomegaly (Figure 3c) with impaired liver function tests (Table 2).

In addition, the patient developed renal tubular acidosis and rickets (Figure 3c). His height and weight were 110.30 cm and 21.4 kg, respectively (Figure 3b). Full pituitary hormonal tests were in the normal range except for the IGF-1 test, which was low (Table 2). The GH stimulation test showed growth hormone deficiency upon which GH 0.5 mg was prescribed daily. Additionally, the patient received multiple medications for his electrolyte imbalance (sodium bicarbonate, potassium, phosphorous, and vitamin D). 

The patient’s WGS revealed a homozygous mutation (c.1093C>T in exon 9, R365X (NM_000340)) in the *SLC2A2* gene, and both parents were carriers. The mutation was confirmed by Sanger sequencing of DNA (Figure 4). Appendix A shows the expected truncated GLUT2 topology for the patient.

### 3.2. GLUT2 Expression in PBMCs

Our study was the first to investigate GLUT2 expression in PBMCs. PBMCs were isolated from patients and healthy controls with no family history of dysglycemia using Ficoll. The expression of GLUT2 protein in each cell type of PBMCs was tested using flow cytometry. The gating strategy used for sorting live PBMCs is illustrated in Appendix A. GLUT2 expression was minimal in CD4+, CD8+, CD19+, and CD14+ populations in PBMCs extracted from healthy controls (no dysglycemia) (Figure 5a). It is noteworthy that the high expression of GLUT2 in CD8+ cells and its expression in CD4+, CD19+, and CD14+ cells was observed in a healthy control who recently received a COVID19 vaccine (Figure 5b). Therefore, these data propose that GLUT2 is expressed in PBMCs and is activated by an immune response. For a better understanding of the molecular mechanisms of dysglycemia in FBS patients, we decided to extract the PBMCs from the patient with the intronic mutation of *SLC2A2* (C. 613-7T>G, IVS 5-7T>G) and from the patient with the exonic mutation of *SLC2A2* (c.1093C>T in exon 9, R365X).

### 3.3. Expression of GLUT2 in Patient and Control PBMCs

Protein lysates were prepared and analyzed by Western blotting to determine the expression of the GLUT2 protein. There was no difference in GLUT2 expression between the patient with the intronic mutation and the control (Figure 6b). However, GLUT2 was overexpressed in the patient with the exonic mutation in comparison to the control (Figure 6a). Two protein bands were visible in the sample obtained from the exonic patient. Two different GLUT2 specific antibodies were used to confirm this result. Although surprising, the presence of full-length GLUT2 in the patient can be explained by GLUT2 mRNA being differentially spliced in the patient, removing the stop codon. There was also a possibility that some type of translational mechanism was “silencing” the stop codon [22].

### 3.4. Glucose Uptake Activity Using Patient PBMCs

To study the effect of GLUT2 mutations on glucose uptake activity, 10 mM 2-DG was used. Remarkably, the 2-DG uptake was significantly decreased in the patient with the exonic mutation in comparison to the age- and gender-matched controls (Figure 7a). In contrast, only a slight decrease (non-significant) in the glucose uptake activity was observed in the patient with the intronic mutation (Figure 7b). 

### 3.5. Expression of of Alternative Glucose Transporters in Patient PBMCs

We investigated the effect of *SLC2A2* mutation on the expression of other glucose transporters. The gene expression patterns of GLUT1 and GLUT3 were assessed using qRT-PCR. We generated equal amounts of cDNA from normalized good-quality RNA extracted from patient PBMCs. The patient with the exonic mutation showed an increase in the expressions of both GLUT1 and GLUT3 in comparison to the healthy control (Figure 8a). However, the patient with the intronic mutation showed a decrease in the expressions of GLUT1 and GLUT3 (Figure 8b). These results suggest the presence of a compensatory mechanism for GLUT2 dysfunction in the patient with exonic mutation, manifested by the activation of expression and function of alternative glucose transporters. However, despite this enhanced expressions of GLUT-1 and -3, the glucose transport activity remained significantly reduced in the patient PBMCs, suggesting a vital role of GLUT2 in glucose transport activity in these cells.

### 3.6. RNA-Seq Analysis of Patient PBMCs

RNA was extracted for RNA-Seq analysis to investigate the differentially regulated genes (DEGs) and the corresponding molecular pathways in patients as compared to their controls. Both patients had DEGs compared to their controls. However, these expression patterns were not related to dysglycemia phenotypes, except for the upregulation of RETN (resistin) in the patient (db-bl-0008) with the intronic mutation (Appendix A), and the upregulation of ENPP1 in the patient (db-bl-1538) with the exonic mutation (Appendix A). Both genes are implicated in insulin resistance and type 2 diabetes mellitus. Both patients displayed DEGs in pathways not associated with dysglycemia (Appendix A). We concluded that the C-terminal portion of GLUT2 is important for its glucose transport activity and that GLUT2 dysfunction is the underlying cause behind dysglycemia in the patient with the exonic mutation. However, we cannot exclude another possible mechanism responsible for dysglycemia in the patient with the intronic mutation.

### 3.7. miRNA Analysis for the Patient with an Intronic Mutation

Since the intronic mutation in *SLC2A2* did not influence the activity of GLUT2, we undertook further investigations to understand the underlying molecular mechanisms of dysglycemia in FBS. We ran Nanostring miRNA panel v3b covering ~800 miRNAs in the patient, mother, and aged- and gender-matching healthy controls with no family history of dysglycemia. We noticed a higher degree of correlation between the patient and the gender- and age-matched healthy control, rather than between the patient and the mother, suggesting that the miRNA expression profile might be influenced further by the age and gender rather than relatedness (Appendix A). The unsupervised hierarchical clustering revealed 123 miRs expressed specifically in the patient sample (Appendix A). The function of these miRNAs was interrogated by using Ingenuity Pathway (IPA) analysis software, which returned 118 mapped miRs (Appendix A). Here, we reported 30 miRNAs with the highest number of count differences in the patient in comparison to the controls (Figure 9). We observed that 14 of them were correlated with T1DM: 10 miRNAs (miR-199a, miR-25-3p, miR-93-5p, miR-19b-3p, miR-107, miR-24-3p. miR-18a-5p, miR-125b-5p, miR-324-5p, miR-331-3p, and hsa-miR-143-3p) were overexpressed in the control as compared to the patient, and 3 miRNAs (miR-144-3p, let-7e-5p, hsa-miR-29a-3p) were significantly overexpressed in the patient in comparison to the control. Molecular networks, including molecules inferred from previous studies, were generated by IPA functional analysis software (Appendix A). The molecular networks were given a score based on the number of molecules represented in the study dataset as compared to the literature. Network 1 (score 33) included the genes and miRNAs implicated in organismal injury and abnormalities, skeletal and muscular system development and function, and tissue morphology. The miR-144 family integrated into network 1. Network 2 (score 31) included insulin and other genes and miRNAs implicated in glomerular injury, inflammatory disease, inflammatory response, and included miR-29 and let-7 families. These results suggest that dysglycemia in the patient with intronic mutation might be associated with the deregulation of miRNAs involved in insulin production and secretion in beta cells.

### 3.8. Expressions of Genes Regulated by miR-29a-3b

Using the RNASeq results, we were interested in investigating if miR-29a-3b overexpression affected the expression of genes involved in insulin production and secretion. We observed that *CAV2, SLC16A1*, *PIK3R1*, and *SLC2A4* were downregulated in the patient in comparison to the control (Appendix A).

## 4. Discussion

FBS patients develop different patterns of dysglycemia, ranging from fasting hypoglycemia, postprandial hyperglycemia, glucose intolerance, to diabetes mellitus [3]. Compound heterozygous or homozygous mutations in *SLC2A2* are the only identified gene mutations implicated with FBS. However, the pattern of dysglycemia does not correlate with the mutation type [3]. The mechanisms involved in the development of dysglycemia in FBS patients are not well characterized. We hypothesized that classic disturbances in GLUT2 structure and/or function was associated with dysglycemia in FBS. 

Glucose uptake activity in peripheral blood mononuclear cells (PBMCs) is vital and plays a key role in inflammation and immune response [23,24,25]. GLUT2 proteins were not expected to be found in blood cells, and they were not present in the plasma using mass spectrometry (https://www.proteinatlas.org/ENSG00000163581-SLC2A2/blood (accessed on 10 June 2021). Fu et al. reported a difference in the expression and immune action of GLUT1, three and five in resting and activated human macrophages, monocytes, and lymphocytes [26]. In addition, Palmer et al. reported that the expression of GLUT1 is significantly increased in pro-inflammatory monocytes from HIV+ persons in comparison to HIV- controls [27]. A recent study showed a significant increase in the expression of GLUT4 in the PBMCs extracted from athletes compared to sedentary participants [28]. Moreover, Haas et al. proved, using a mouse model, an increase in the expression level of GLUTs (1, 2, 3, and 4) in CD4+ cells activated with CXCL100 [29]. 

We were interested in examining the expression of GLUT2 in human PBMCs to study the impact of *SLC2A2* mutations on glucose uptake activity. We successfully detected for the first time the expression of GLUT2 in human PBMCs using qRT-PCR. Then, we studied the expression of GLUT2 in different cell types of human PBMCs (T lymphocytes (CD4+ and CD8+), B lymphocytes (CD19+), and monocytes (CD14+)). We observed a very low expression of GLUT2 in different cell types of PBMCs extracted from the healthy control (no dysglycemia) (Figure 5). However, the expression of GLUT2 was upregulated in all activated PBMCs cell types of a healthy control who recently received the COVID-19 vaccine.

Based on the results that we generated from the patient’s PBMCs, we suggest that the exonic mutation in GLUT2 affects its glucose transport activity and that this explains the dysglycemia observed in this FBS patient. This result is consistent with our recent works confirming the role of the last loops (loops 9–12) of GLUT2 in glucose transport activity [30]. However, it is still unclear if and how the intronic mutation in GLUT2 impacts on glucose metabolism. In an effort to address this question, we used the Nanostring miRNA panel v3b to investigate if the intronic mutation affected the expression of miRNAs correlated with dysglycemia. The molecular analyses showed that the miRNA expression profiles of the patient were more similar to that of the healthy control than to that of the mother (Appendix A). We identified 123 miRNAs that were expressed specifically in the patient, and the subsequent IPA analysis identified 118 mapped miRNAs (Appendix A). The highest 30 counts of difference in the expression in the patient in comparison to the control were presented, and 14 miRNAs (miR-199a, miR-25-3p, miR-93-5p, miR-19b-3p, miR-107, miR-24-3p. miR-18a-5p, miR-125b-5p, miR-324-5p, miR-331-3p, miR-144-3p, let-7e-5p, hsa-miR-29a-3p, and hsa-miR-143-3p) were correlated with type 1 DM (Figure 9). 

Interestingly, we observed the expression of hsa-miR-29a-3p was significantly increased in the patient in comparison to the healthy control. Aghaei et al. reviewed the miR-29 family and its association with insulin secretion and identified three separate mechanisms; either by the direct targeting of pancreatic p85α or Stx-1a to influence insulin signaling or fusion of insulin granule with the membrane, respectively, or by targeting hepatic p85α to activate gluconeogenesis [31]. 

Furthermore, one study demonstrated that miR-29a inhibited glucose-stimulated insulin secretion (GSIS) and cell proliferation in MIN6 cells via a negative effect on Cdc42/β-Catenin signaling [32]. It has also been suggested that miR-29a inhibits GSIS by targeting syntaxin-1 and Mct1, as well as insulin signaling by targeting INSIG1, CAV2, and PIK3R1 [33,34]. Furthermore, Zhou et al. reported that miR-29a is also implicated in insulin resistance by decreasing ATP production, GLUT4 expression, and glucose uptake through targeting PPARδ [35]. Hromadnikova et al. suggested that hsa-miR-29a-3p could play a role in heart disease and diabetes mellitus [36]. In addition, a recent study showed that miR-29 is involved in inflammation and diabetes mellitus through the downregulation of TRAF3 [37]. Interestingly, we observed that the expression of *CAV2, SLC16A1, PIK3R1,* and *SLC2A4* were downregulated in the patient carrying the intronic GLUT2 mutation in comparison to control (Appendix A). TargetScan (prediction of miRNAs target website) predicted an interaction between miR-29a-3b and position 1341–1347 in the human 3’UTR of *SLC2A2* (ENST00000314251.3). However, miRecords did not predict any interaction. 

In addition, two other miRNAs, hsa-miR-144-3p and hsa-let-7e-5p, were also significantly increased in the patient in comparison to the healthy control. Shen et al. showed hsa-miR-144-3p is associated with adipogenesis by promoting C/EBPα activity [38]. Demirsoy et al. reported that the expression of hsa-let-7e-5p was significantly downregulated in patients with T2D after receiving metformin therapy [39]. TargetScan proposed an interaction between has-let-7e-5p and position 878–884 in the 3′UTR of *SLC2A2*. However, miRecords did not predict any interaction.

Moreover, we observed that another set of 11 miRNAs (miR-199a, miR-25-3p, miR-93-5p, miR-19b-3p, miR-107, miR-24-3p. miR-18a-5p, miR-125b-5p, miR-324-5p, miR-331-3p miR-199a, miR-25-3p, miR-93-5p, miR-19b-3p, miR-107, miR-24-3p. miR-18a-5p, miR-125b-5p, miR-324-5p, miR-331-3p, and hsa-miR-143-3p) was overexpressed in the patient in comparison to the control. Jordan et al. reported that miR-143 impaired the capability of insulin to stimulate AKT activation and glucose homeostasis by the downregulation of oxysterol-binding-protein-related protein 8 (ORP8) [40]. Guo et al. reported that the expression of miR-324-5p was elevated in patients with hyperlipidemia and hyperglycemia due to the suppression of ROCK1 [41]. Yu et al. reported that the overexpression of miR-125b-5p improved the function of the pancreatic β cell through suppression of DACT1 [42]. Tavano et al. reported that miR-18a-5p was overexpressed in both pancreatic and non-pancreatic cancers with a recent onset of diabetes in comparison to the healthy controls [43]. Xu et al. reported that the overexpression of miR-125a-5p enhanced hepatic glucose and lipid metabolism (decreasing lipid and glucose levels and increasing glycogen storage) in type 2 diabetes through the inhibition of STAT3 expression [44]. Overall, our miRNA analysis clearly demonstrated that the patient carrying the intronic GLUT2 mutation displayed a different miRNA profile compared to the healthy controls. Interestingly, many of the overexpressed miRNAs have been linked to cardiometabolic disease. However, if and how the mutation in GLUT2 is linked to the dysregulated expression of these miRNAs remains to be explored.

One limitation of this study was that we only studied two patients, each with different mutations in the GLUT2 gene. Thus, we were unable to rule out other possible mechanisms of dysglycemia in FBS patients. In addition, the limited amount of patient samples available for this study prevented us from fully characterizing the expressions of GLUT2 and GLUT4 in the PBMCs from the patient carrying the exonic mutation. Future studies should address this by using CRISPR/Cas9 gene editing to mimic the exonic mutation in cells important for organismal glucose homeostasis. Further functional analyses are also required to prove the causality of the increased expression of miR-29a-3p in the patient carrying the intronic GLUT2 mutation.

## 5. Conclusions

Our study confirmed that homozygous *SLC2A2* mutations were involved in the development of dysglycemia in FBS, either by a direct effect on GLUT2 expression and/or activity or by an indirect effect on other molecules involved in glucose homeostasis.

## Figures and Tables

**Figure 1 biomedicines-10-02114-f001:**
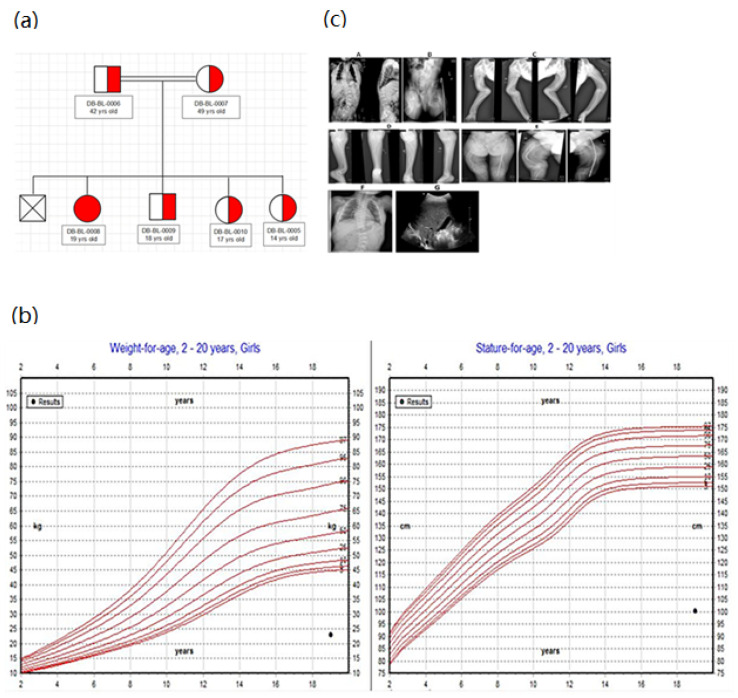
**Clinical characteristics of case 1:** (**a**) family pedigree (patient descends from first-degree cousins). (**b**) Growth charts. According to WHO length chart: a patient has short stature (dot) and is underweight (dot). (**c**) Radiological findings (rickets and bone deformities) (**A–F**) and hepatomegaly (**G**).

**Figure 2 biomedicines-10-02114-f002:**
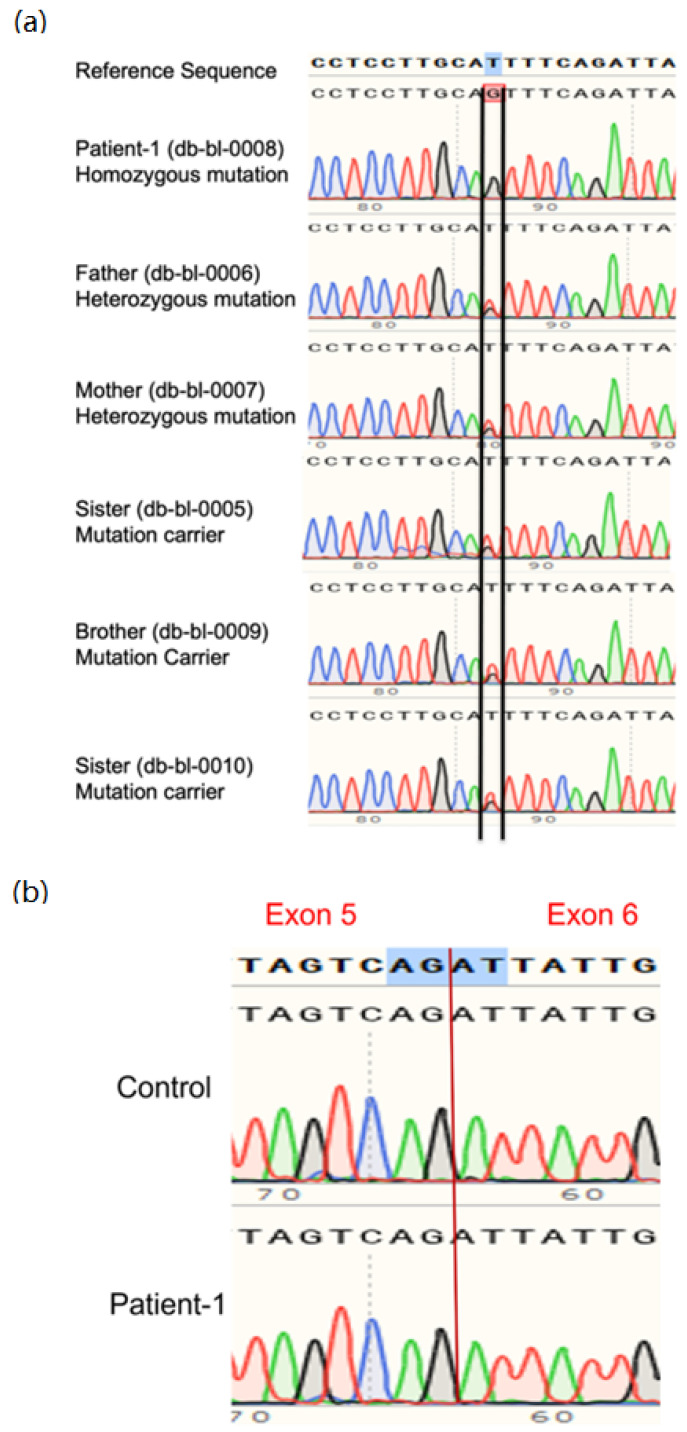
**Genetic analysis for case 1 and family.** (**a**) Sanger sequencing of DNA of the patient showed a novel homozygous mutation of *SLC2A2* (C. 613-7T>GIVS 5-7T>G) expected to effect a splice site between exons 5 and 6. (**b**) cDNA sequencing demonstrates that the splicing of exons 5 and 6 are unaffected by the intronic mutation. Parents are carriers of the mutation.

**Figure 3 biomedicines-10-02114-f003:**
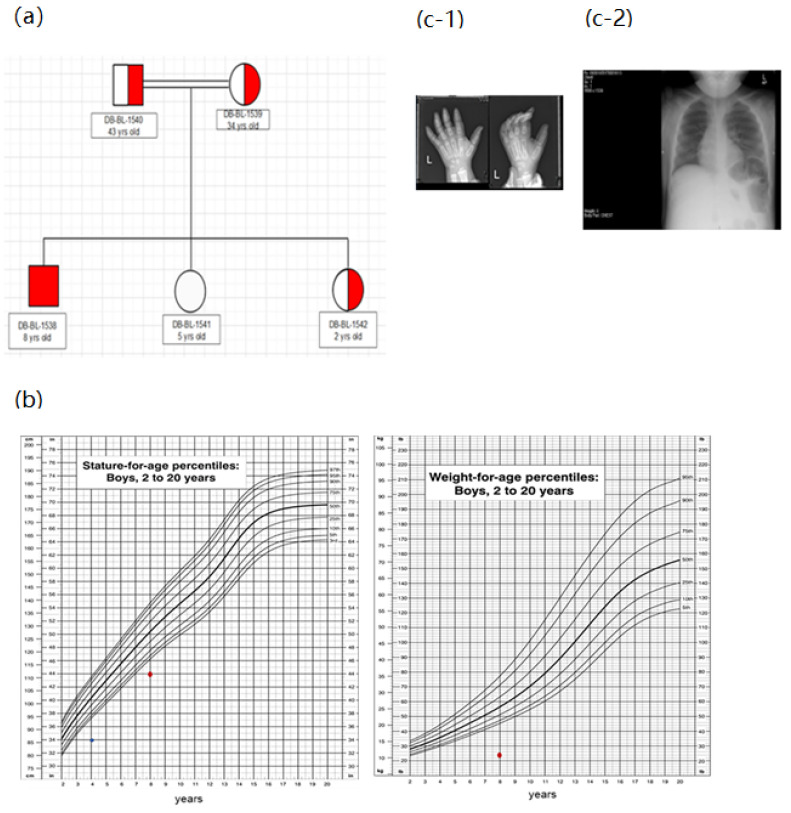
**Clinical characteristics of case 2:** (**a**) Family pedigree (patient descends from first-degree cousins). (**b**). Growth charts. According to CDC length chart, patient has short stature (dot) and is underweight (dot). (**c**) Radiological findings demonstrating the development of rickets (**c-1**) and hepatomegaly (**c-2**).

**Figure 4 biomedicines-10-02114-f004:**
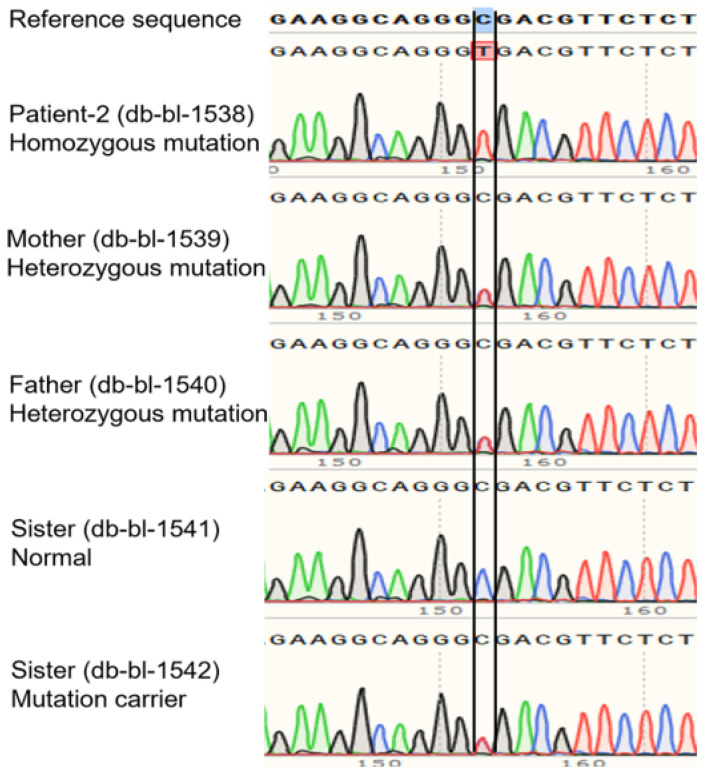
**Genetic analysis for case 2 and family.** Sanger sequencing of DNA shows a homozygous mutation of *SLC2A2* (c.1093C>T in exon 9, R365X) in the patient. Both parents are carriers of the mutation.

**Figure 5 biomedicines-10-02114-f005:**
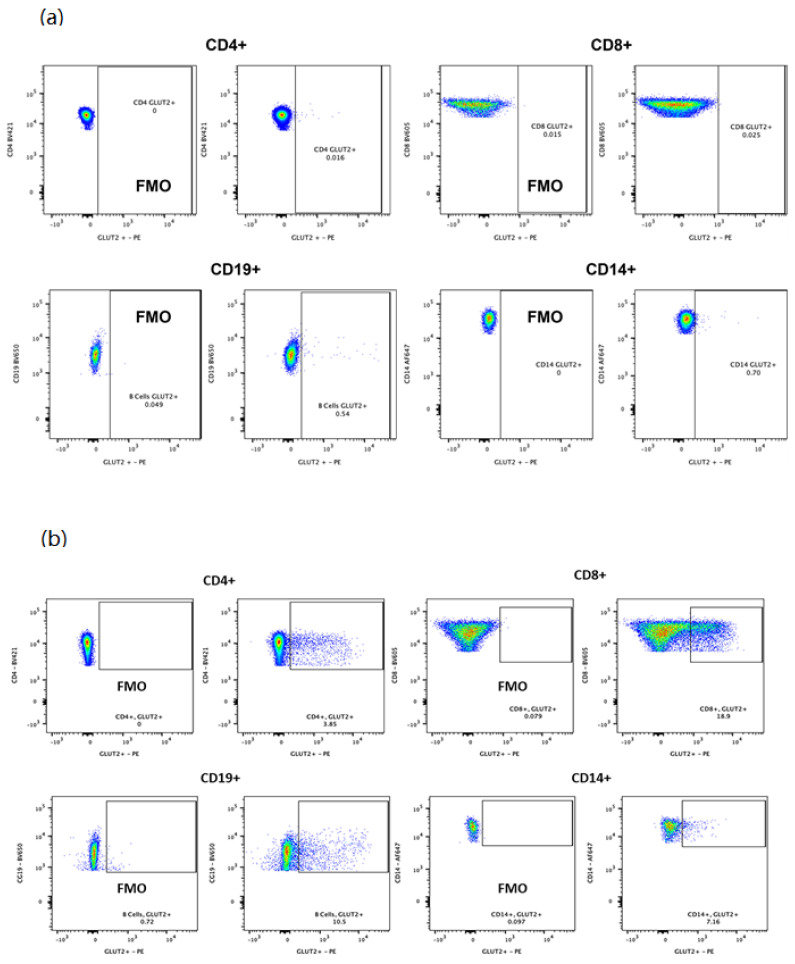
**Flow cytometry to assess GLUT2 expression in each cell type in healthy PBMCs.** (**a**) GLUT2 expression was very low in all cell populations in PBMCs extracted from the healthy control. (**b**) However, the expression of GLUT2 was increased in specific cell populations in PBMCs extracted from an immune-activated control.

**Figure 6 biomedicines-10-02114-f006:**
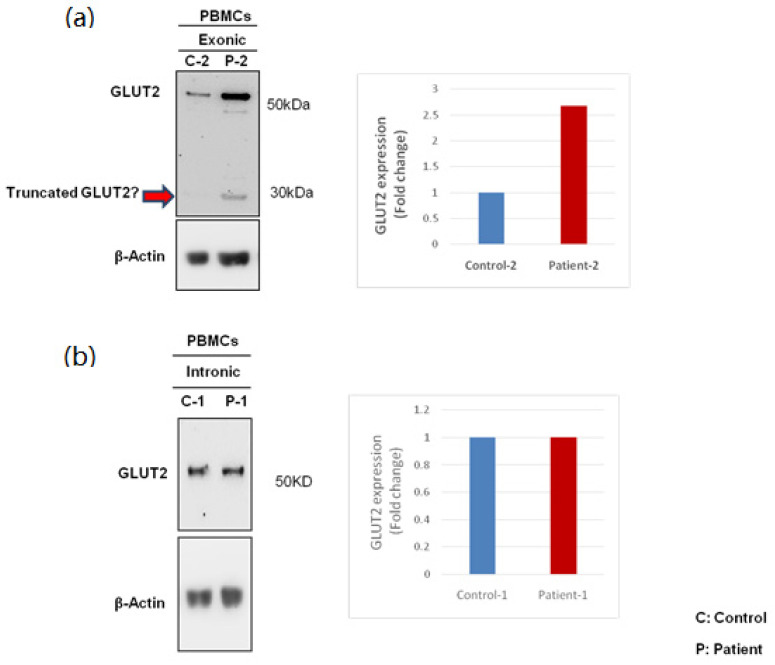
**Western blotting to assess the expression of GLUT2 in patient and control PBMCs.** (**a**) The expression of GLUT2 was increased in the patient with the exonic mutation. (**b**) The expression of GLUT2 was similar in the patient with the intronic mutation in comparison to control. A smaller band was detected by the GLUT2 antibody in the sample obtained from the patient carrying the exonic mutation, indicated by the arrow. The quantification (fold change) of GLUT2 in the samples are presented to the right in both (**a**,**b**).

**Figure 7 biomedicines-10-02114-f007:**
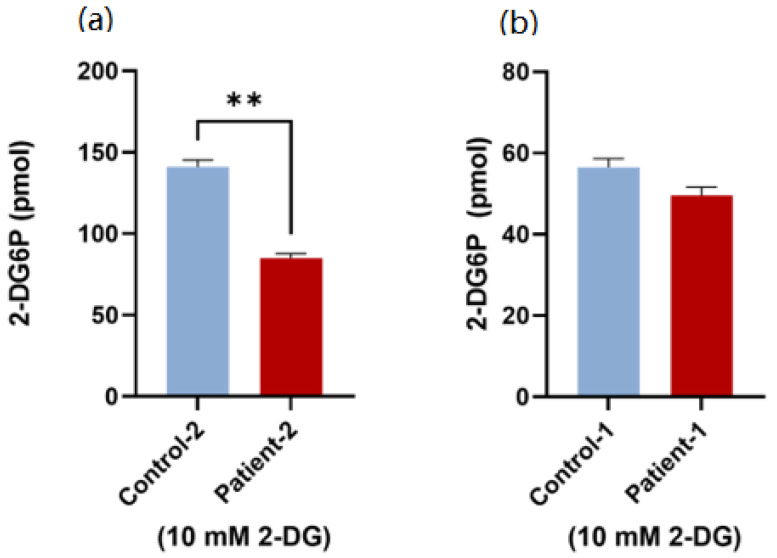
**Glucose uptake test using patient PBMCs.** (**a**) PBMCs from the patient with the exonic mutation had significantly decreased glucose uptake activity in comparison to the control. (**b**) PBMCs from the patient with intronic mutation displayed the same glucose uptake activity as the control. *p*-value was calculated using two-tailed *t*-test and presented with a ‘‘*’’ in the graph. ** *p* ≤ 0.01. Error bar is the average reading of three independent experiments.

**Figure 8 biomedicines-10-02114-f008:**
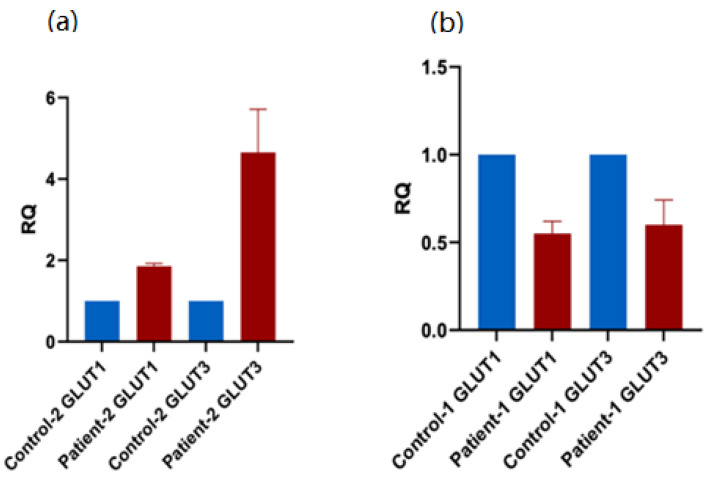
**qRT-PCR to assess the expression of glucose transporters in patient PBMCs.** (**a**) The mRNA levels of both GLUT1 and GLUT3 are elevated in PBMCs obtained from the patient carrying the exonic mutation. (**b**) In contrast, the mRNA levels of GLUT1 and GLUT3 are reduced in PBMCs obtained from the patient carrying the intronic GLUT2 mutation. Error bar is the average of technical replicates.

**Figure 9 biomedicines-10-02114-f009:**
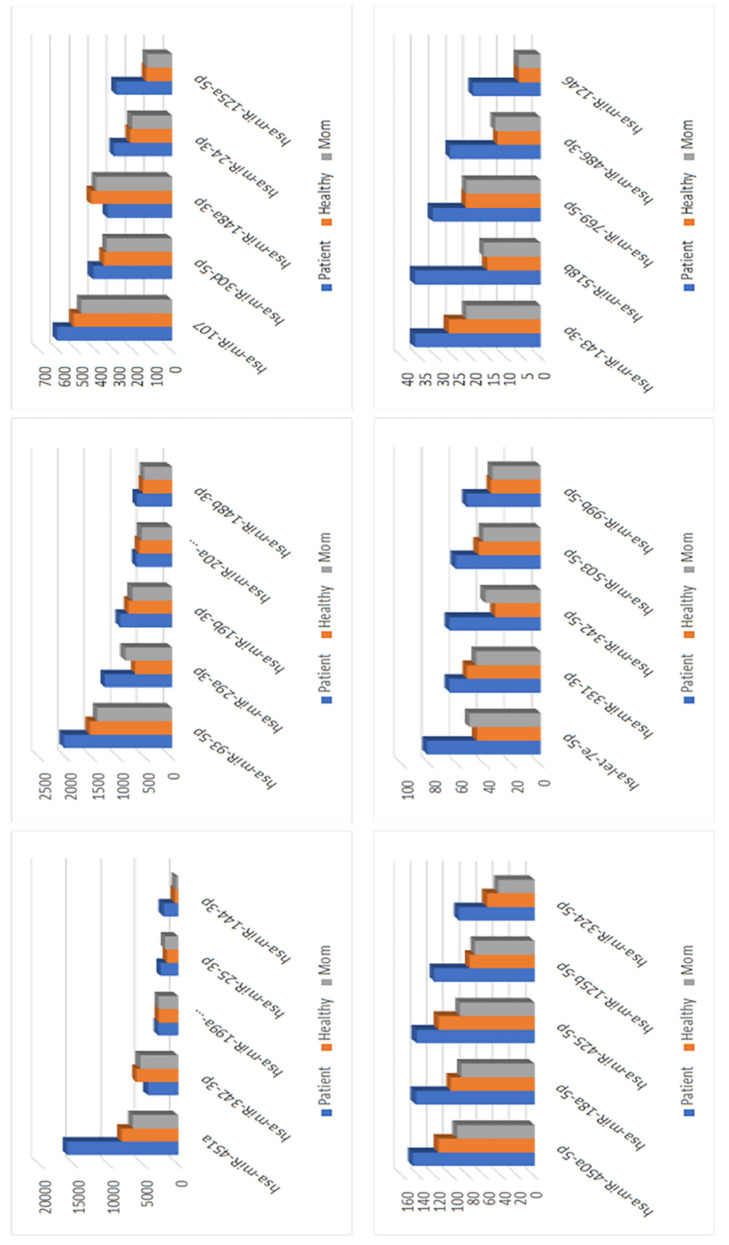
The 30 selected miRNAs with the highest number of counts difference in the patient with the intronic mutation in comparison to the healthy control. Fourteen miRNAs expressed in type 1 diabetes mellitus are upregulated in the patient in comparison to control, and three of these (miR-144-3p, let-7e-5p, hsa-miR-29a-3p) are significantly overexpressed.

**Table 1 biomedicines-10-02114-t001:** Summary of biochemical tests for case 1.

Investigation	Test Value	Normal Range
Electrolyte level		
Serum phosphorus (mmol/L)	0.43	0.93–1.64
Serum calcium (mmol/L)	2.15	2.2–2.7
Serum Sodium (mmol/L)	136	134–146
Serum Potassium (mmol/L)	4	3.5–5.0
Liver function test		
Alanine amino transferase (ALT) (IU/L)	56	8–22
Aspartate transaminase (AST) (IU/L)	74	0–30
Alkaline phosphatase (IU/L)	1108	48–95
Blood glucose test		
Fasting glucose (mmol/L)	2.5	3.5–5.5
2 h glucose tolerance (mmol/L)	28.2	7.8–11.1
C peptide (ng/mL)	0.68	0.78–5.19
Hemoglobin A1c (HbA1c)%	6.4	4.8–6.0
Diabetes mellitus type 1 evaluation **(GAD65, ICA, ZnT8, IAA)**	Negative	-
Growth hormone test		
Thyroid stimulating hormone (TSH) (mIU/L)	3.82	0.4–4.0
Lipid test		
Cholesterol (mmol/L)	6.53	<5.18
Triglyceride (mmol/L)	1.3	<1.7
High density lipoprotein (HDL-C) (mmol/L)	0.9	>1.17
Low density lipoprotein LDL (mmol/L)	4.72	<2.6
Kidney test		
Blood urea nitrogen (BUN) (mmol/L)	1.8	1.9–6.7
Creatinine (µmol/L)	35	54–95
Urinalysis	Generalized aminoaciduria ++, phosphaturia, glucosuria (+2)	

**Table 2 biomedicines-10-02114-t002:** Summary of biochemical tests for case 2.

Investigation	Test Value	Normal Range
Electrolyte level		
Serum phosphorus (mmol/L)	0.79	0.93–1.64
Serum calcium (mmol/L)	2.17	2.2–2.7
Serum Sodium (mmol/L)	135	134–146
Serum potassium (mmol/L)	3.3	3.5–5.0
Liver function test		
Alanine amino transferase (ALT) (IU/L)	30	8–22
Aspartate transaminase (AST) (IU/L)	40	0–30
Alkaline phosphatase (IU/L)	388	48–95
Blood glucose test		
Fasting glucose (mmol/L)	3.2	3.5–5.5
2 h post-feed glucose (mmol/L)	18.3	3.5–5.5
Insulin (pmol/L)	6	18–48
Hemoglobin A1c (HbA1c)%	5.7	4–5.6
Diabetes mellitus type 1 evaluation **(GAD65, Insulin, ZnT8, IA-2)**	Negative	-
Growth hormone test		
Thyroid stimulating hormone (TSH) (mIU/L)	1.72	0.4–4.0
Insulin-like growth factor-1 (IGF-1) (nmol/L)	7.8	8.2–30.8
Parathyroid hormone (PTH intact)	10.3	2.0–6.8
Kidney test		
Blood urea nitrogen (BUN) (mmol/L)	3	2.5–7.1
Creatinine (µmol/L)	34	60–110
Urinalysis	Proteinuria (+1), glucosuria (+3)	

## Data Availability

The RNA-Seq and Nanostring assay data have been deposited in the NCBI GEO (GSE198678).

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
