# Peer review of "Understanding the Role of GLUT2 in Dysglycemia Associated with Fanconi–Bickel Syndrome"

_biomedicines, 2022, doi:10.3390/biomedicines10092114_

Round 1
Reviewer 1 Report
The manuscript is well written. the author represented their findings in a logical and scientific way. I enjoy to reading it. I accept the present form without any corrections.
Author Response
Please see the attached response report

Reviewer 2 Report
In this manuscript, the authors describe two patients with FBS who have different SLC2A2 gene mutations and focus on trying to understand the mechanisms of dysglycemia. Although the topic of research is not very novel, experiments were performed appropriately and well presented. I recommend acceptance after minor language and text editing.
Author Response
Please see the attached response report

Reviewer 3 Report
In the present manuscript written by Sharari et al., the authors described that SLC2A2 mutations cause dysglycemia in patients with Fanconi-Bickel Syndrome (FBS) either by a direct effect on GLUT2 expression or activity or, indirectly, by the dysregulated expression of miRNAs implicated in glucose homeostasis. The conducted research is very interesting.
The manuscript is well-written, presented figures are of good quality and limitations of the study have been taken into account. However, below there are points, which should be taken into account:
1. In material and methods sections authors should provide subsection “Statistical analysis” and describe the statistical tests used.
2. In material and method section in Western Blot subsection authors should provide the method used to determine the total protein.
3. In results section in Figure 6, there should be beta-actin (β-actin) instead of actin.
4. The authors should provide the explanation of all abbreviations in the manuscript (e.g. GLUT2 line 55), where they are first used.
5. The authors should describe the aim of the study in more detail.
6. In the Tables 1 and 2 not all abbreviations are explained.
7. The authors conducted studies on two relatively different patients (case 1 – 19-yeras old female, case 2 – 8-years old boy) with different mutations in the GLUT2 gene, as described in the manuscript. Perhaps it is worth considering whether factors such as age, gender and sexual maturity had an effect on GLUT2 expression and glucose intake.
Author Response
We thank the reviewer for the comments. Please see the attached response report

This manuscript is a resubmission of an earlier submission. The following is a list of the peer review reports and author responses from that submission.